# Understanding Proton Magnetic Resonance Spectroscopy Neurochemical Changes Using Alzheimer’s Disease Biofluid, PET, Postmortem Pathology Biomarkers, and APOE Genotype

**DOI:** 10.3390/ijms251810064

**Published:** 2024-09-19

**Authors:** Firat Kara, Kejal Kantarci

**Affiliations:** Department of Radiology, Mayo Clinic, Rochester, MN 55905, USA; kantarci.kejal@mayo.edu

**Keywords:** Alzheimer’s disease, mild cognitive impairment, magnetic resonance spectroscopy, amyloid, tau, biofluid biomarkers, apolipoprotein *e*4

## Abstract

In vivo proton (^1^H) magnetic resonance spectroscopy (MRS) is a powerful non-invasive method that can measure Alzheimer’s disease (AD)-related neuropathological alterations at the molecular level. AD biomarkers include amyloid-beta (Aβ) plaques and hyperphosphorylated tau neurofibrillary tangles. These biomarkers can be detected via postmortem analysis but also in living individuals through positron emission tomography (PET) or biofluid biomarkers of Aβ and tau. This review offers an overview of biochemical abnormalities detected by ^1^H MRS within the biologically defined AD spectrum. It includes a summary of earlier studies that explored the association of ^1^H MRS metabolites with biofluid, PET, and postmortem AD biomarkers and examined how apolipoprotein *e*4 allele carrier status influences brain biochemistry. Studying these associations is crucial for understanding how AD pathology affects brain homeostasis throughout the AD continuum and may eventually facilitate the development of potential novel therapeutic approaches.

## 1. Introduction

Alzheimer’s disease (AD) is an irreversible neurodegenerative disorder accounting for 60–80% of all dementia cases. More than 55 million people live with AD and other dementias, and this number is expected to rise [1]. AD dementia imposes a significant burden on the healthcare system and primary caregivers. In 2016, dementia was globally the fifth-leading cause of death, contributing to an estimated 2.4 million deaths (95% UI 2.1–2.8 million), following ischaemic heart disease, chronic obstructive pulmonary disease, intracerebral hemorrhage, and ischaemic stroke [2,3]. In terms of Disability-Adjusted Life Years (DALYs), dementia caused 28.8 million (95% UI 24.5–34.0) DALYs globally, making it the 23rd largest contributor to DALYs, rising significantly from 41st place in 1990 [2,3]. This increase is largely attributable to the aging population, particularly in low- and middle-income countries, where demographic changes are accelerating faster than in high-income nations [2,3,4]. Patients with AD dementia experience a progressive decline in both cognitive and physical abilities, leading to substantial challenges in daily functioning and quality of life. Recent progress in disease-modifying therapies, such as monoclonal antibodies, has emphasized the need for early and precise diagnosis of AD’s underlying pathology through surrogate biomarkers [5]. Detecting AD pathology at the earliest stage, before significant brain damage occurs, is essential for timely intervention, which can improve outcomes. Proton (^1^H) magnetic resonance spectroscopy (MRS) is a non-invasive method to detect changes in brain biochemistry linked to molecular-level pathological processes related to AD. The integration of ^1^H MRS with other biomarkers acquired through positron emission tomography (PET) and biofluid analysis enhances the ability to diagnose AD at an early stage and monitor the impact of interventions, potentially leading to improved patient outcomes. 

^1^H MRS offers the advantage of monitoring biochemical alterations over time, making it a valuable tool for tracking disease progression as a prognostic tool and the efficacy of therapeutic interventions. Monitoring metabolite changes during the AD continuum provides a more detailed picture of the disease process, allowing for the identification of specific biochemical pathways affected by AD pathology. A variety of metabolites, which reflect different physiological functions, can be measured in the brain using ^1^H MRS. For example, alterations in the total N-acetylaspartate (tNAA: NAA plus N-acetyl aspartylglutamate, NAAG) signal is associated with neuronal integrity, myo-inositol (mIns) reflects glial activity, glutamate (Glu) and gamma-aminobutyric acid (GABA) represent excitatory and inhibitory neurotransmission, respectively, while total choline (choline, Cho; phosphocholine, PCho; glycerophosphocholine, GPC) is associated with cell membrane turnover, and total creatine (tCr = creatine, Cr plus phosphocreatine, PCr) is linked to energy metabolism. Given the varying precision in naming some of the ^1^H MRS metabolites, NAA, Cr, and Cho may sometimes refer to the total sum as explained in the “total” definitions in this review.

The accumulation of amyloid-beta (Aβ) plaques outside the neurons and the accumulation of neurofibrillary tangles composed of hyperphosphorylated tau inside the neurons start more than two decades before the clinical symptoms of AD dementia manifest [6,7,8]. AD neuropathology can be measured through postmortem examination or using biomarkers in living people. The clinical diagnostic criteria for AD dementia have evolved over time. In 2018, the US National Institute on Aging and the Alzheimer’s Association (NIA-AA) committee proposed the biological definition and diagnosis of AD as a research framework by using biomarkers acquired from living people during the AD continuum [9]. The NIA-AA framework has enabled a purely biological definition of AD by categorizing individuals based on biomarker evidence of AD pathology using amyloid-tau-neurodegeneration (ATN) [9]. In ATN classification, “A” refers to the Aβ proteinopathy pathway, “T” to the tau proteinopathy pathway, and “N” to neurodegeneration. This biomarker classification scheme has recently been revised [10]. Revised criteria for the diagnosis and staging of AD have introduced three new biomarker categories: inflammatory/immune mechanisms, vascular brain injury, and alpha (α)-synucleinopathy [10]. Growing evidence suggests that AD often co-occurs with multiple pathologies in older adults [10]. Additional pathologies associated with AD are gaining increased attention in efforts to better understand its pathological heterogeneity. For instance, the presynaptic protein α-synuclein, primarily linked to Parkinson’s disease, dementia with Lewy bodies, and multiple system atrophy, has been detected in over half of postmortem-confirmed AD brains [11,12,13]. Emerging areas of interest include astrocyte calcium dysregulation [14] and calcium’s interaction with α-synuclein [15], as well as inflammation-related proteins like pro-inflammatory S100A9 [16], which are being explored to gain a deeper understanding of AD’s underlying pathological heterogeneity [17]. While the newly revised biological disease staging using PET and biofluid markers will continue to increase our understanding of the disease progression, ^1^H MRS biomarkers have the potential to add a crucial neurochemical dimension to increase our understanding regarding neuropathological alterations in the AD continuum and enhance the accuracy of early AD diagnosis. Unlike biofluid biomarkers, which provide systemic-level insights into changes in AD biomarkers such as Aβ and tau, ^1^H MRS allows for regionally specific biochemical assessments in the brain. 

The biologically defined AD continuum begins with the appearance of brain pathology in asymptomatic individuals and progresses through stages of increasing pathological burden, eventually leading to clinical symptoms [9,18]. The AD continuum stages include the preclinical stage, where the participants are cognitively unimpaired (with or without amyloid or tau biomarkers); the prodromal stage includes the early symptomatic phenotype (e.g., MCI), and the AD dementia stage refers to the phase where severe cognitive symptoms are present which affect social and daily activities [19]. With recent advances in in vivo biomarker fields (i.e., neuroimaging and biofluid markers), progression of AD pathology in vivo is now possible starting from preclinical stages long before the irreversible brain damage occurs.

The focus of this review was to summarize the relationship between ^1^H MRS metabolites and biomarkers acquired through PET (tau, amyloid), biofluid, and postmortem AD pathology analysis across the AD continuum in the brain (Table 1). There are several excellent older reviews on changes in ^1^H MRS metabolites in the AD continuum for further reading [20,21,22,23,24,25,26,27,28].

## 2. Commonly Studied ^1^H MRS Metabolites in AD 

### 2.1. NAA

NAA is a small molecule that is synthetized from aspartate and acetyl-coenzyme A in the brain. It is a marker of neural health, viability, and synaptic integrity [29]. NAA has a high amplitude signal at 2.01 part per million (ppm), relative to the standard tetramethysilane. The peak also includes minor contributions from other metabolites such as NAAG at 2.04 ppm. These overlapping signals from the acetyl moieties of both molecules at the 2.01–2.04 ppm range (plus lower-amplitude signals from other ppm ranges; see de Graaf, 2007) are ascribed as NAA or total NAA (tNAA: NAA + NAAG) [30,31,32,33,34,35]. NAA is found primarily in neural cells and synthesized in mitochondria [31,32,33]. A lower NAA/Cr ratio has been associated with a loss of synaptic integrity [29]. A reduction in NAA levels (using water as an internal reference or Cr) in patients with AD dementia compared to cognitively unimpaired participants is one of the most frequent findings of ^1^H MRS studies [36,37,38,39,40,41,42,43,44,45,46,47,48,49,50]. 

### 2.2. mIns

The well-resolved spectral peak of mIns is assigned to 3.56 ppm [51,52]. mIns is considered as a glial marker and/or an osmolyte [53]. An increase in mIns was linked with an elevated glial marker on PET (18kDa TSPO). TSPO PET uptake has been associated with neuroinflammation and glial cell activation [54] or the density of inflammatory cells [55]. This association supports the notion that mIns can be a marker of neuroinflammation or the density of inflammatory cells. Many studies consistently demonstrated an increase in mIns (or mIns/Cr) in several gray- and white matter brain regions in people with AD dementia compared to controls [22,36,37,38,39,41,46,47,48,50,53,56,57]. It has been proposed that an increase in the mIns/Cr ratio occurs during the early stages of the disease progression, which is then followed by a decrease in the NAA/Cr ratio and an increase in the Cho/Cr ratio at later stages of the disease [50]. 

**Table 1 ijms-25-10064-t001:** Summary of association studies between ^1^H MRS and other biomarkers including biofluid, PET, and postmortem pathology.

References	Cohort	Magnet Field Strength and Acquisition Parameters	Voxel Locations and Size	Key Findings
[58]	CU (n = 30)	7T, TR = 644, MRSI, FIDLOVS	Posterior cingulate gyrus and precuneus	↑ GABA and ↑ Glu were associated with ↑ Aβ burden on PET (PiB) with a positive effect modification by *APOE e*4 allele.
[59]	AD (11), MCI (8), CU (n = 26)	3T, TR/TE = 2000/30 ms, MRSI, PRESS	Posterior cingulate gyrus, dorsolateral prefrontal cortex	↓Glu/tCr was associated with ↑ tau load on PET with florzolatau in the posterior cingulate gyrus of AD dementia patients. ↑ plasma NfL was associated with MRS metabolites (↓ tNAA/tCr and ↓ Glu/tCr) in the right dorsolateral prefrontal cortex of patients with AD dementia.
[60]	CU (Aβ–and Aβ+) (n = 338), MCI (Aβ+)(n = 90)	3T, TR/TE = 2000/30 ms, single voxel, PRESS	Posterior cingulate cortex/precuneus region	↑ mIns/tCr ratio in the posterior cingulate gyrus was associated with ↑ posterior cingulate gyrus and neocortical meta-ROI Aβ (flutemetamol) and tau (RO948) load on PET only in *APOE e*4 allele carriers. ↑ plasma GFAP was associated with ↑ mIns/tCr (posterior cingulate gyrus) only in *APOE e*4 allele carriers.
[61]	CU women: CSF-Aβ-negative (n = 71); CU-Aβ-positive women (n = 37); MCI (CSF-Aβ-positive) women (n = 12)	3T, TR/TE = 2000/20 ms; TR/TE = 2000/68 ms/; single voxel; PRESS and MEGA-PRESS	Medial frontal cortex	↑ Glx, ↓ GABA, and ↑ mIns/tCr ratio in MCI compared to CU CSF-Aβ42-negative and -positive participants. ↑ Age was associated with ↓ levels of GABA in CU and MCI groups.
[62]	CU (A−T−N−) (37); early AD (A+T+N−) (n = 16); late AD (A+T+N+) (n = 15)	3T, TR/TE = 2000/32 ms; single voxel; PRESS	Posterior cingulate cortex/precuneus region	↓ NAA/Cr in early AD (A+T+N−) and late AD (A+T+N+) compared to controls (A−T−N−; A+T−N−). ↑ mIns/Cr in late AD compared to controls. ↓ NAA/Cr correlated with ↑ global Aβ load (PIB) and tau load (flortaucipir) on PET in whole cohort.
[63]	CU (n = 40)	3T, TR/TE= 3000/30 ms, single voxel; sLASER	Posterior cingulate gyrus (automated VOI prescription)	↑ Tau PET (flortaucipir) in posterior cingulate gyrus correlated with ↓ NAA/tCr and ↓ Glu/tCr.
[64]	CSF Aβ42 positive (n = 111); CSF Aβ42 negative (n = 174);	3T, TR/TE= 3000/30 ms, single voxel; PRESS	Posterior cingulate cortex/precuneus region	Visit 2 (~2.3 years after baseline): ↑ Cho/Cr, ↑ mIns/Cr, ↓ NAA/Cr, and ↓ NAA/mI in CSF-Aβ-positive compared to CSF-Aβ-negative cases. Visit 3 (~4 years after baseline): ↑mIns/Cr, ↓ NAA/Cr, and ↓ NAA/mI in CSF-Aβ-positive compared to CSF-Aβ-negative cases. CSF Aβ positivity at baseline was associated with ↑ mIns/Cr and ↓ NAA/mIns ↑ Rate of change in the MCI Aβ positive for mIns/Cr and NAA/mIns compared to MCI Aβ negative.
[65]	CU younger controls (<60 years) (n = 27); CU older controls (>60 years) (n = 27); AD (>60 years) (n = 25)	3T, TR/TE = 1600/(31–229) ms ms, single voxel, 2D J-PRESS	Posterior cingulate cortex/precuneus region	↑ mIns associated with ↑CSF tau and ↑ CSF p-Tau 181; ↑ GABA associated with ↑CSF p-Tau 181p in AD dementia group.
[66]	Two cohorts: younger age (n = 30) (20–40 years); CU (n = 151): older individuals (60–85 years).	3T, TR/TE = 4000/8.5 ms, single voxel, SPECIAL	Posterior cingulate cortex/precuneus region	↑ mIns, ↑ Cr, ↑ mIns/NAA, ↓ GSH, and ↓ Glu in older participants compared to younger participants.
[67]	CU (n = 289)	1.5T, TR/TE = 2000/25 ms, single voxel, PRESS	Posterior cingulate gyrus	↑ mIns/Cr ratio in participants with two copies of *APOE e*4 allele compared with participants with non-carriers. ↓ The NAA/mIns ratio in participants (*APOE e*4/*e*4) compared with those who were heterozygous for the *APOE e*4 allele and non-carriers.
[68].	CU (n = 15)	3T, TR/TE = 1500/68 ms, single voxel, J-edited spin echo difference method	Posterior cingulate cortex/precuneus region	↓ GSH was associated with↑ the temporal and parietal Aβ load on PET with PiB.
[69]	aMCI (n = 14); CU (n = 32)	3T, TR/TE= 3000/30 ms, single voxel, sLASER	Posterior cingulate gyrus	↑ Global cortical Aβ load (PiB) on PET correlated with ↓ Glu/mIns ratio in the entire cohort.
[70]	CU older adults (n = 594) ^c^	3T, TR/TE= 2000/30 ms, single voxel, PRESS	Posterior cingulate gyrus	↓ NAA/mIns and ↑ mIns/Cr at baseline were associated with ↑ rate of Aβ deposition on serial PIB PET.
[71]	CU CSF Aβ42 negative (n = 156); CU CSF Aβ42 positive (n = 49), MCI CSF Aβ42 positive (n = 88)	3T, TR/TE= 2000/30 ms, single voxel, PRESS	Posterior cingulate/precuneus	↑ mIns/Cr, ↑ Cho/Cr, and ↓ NAA/Cr in MCI (CSF Aβ42 positive) compared to CU (CSF Aβ42 negative). ↑ mIns/Cr in CU (CSF Aβ42 positive) compared to CU (CSF Aβ42 negative). ↑ mIns/Cr in *APOE e*4 allele carrier CU (CSF Aβ42 negative) compared to non-*e*4-carrier CU (CSF Aβ42 negative). ↑ mIns/Cr and ↑ Cho/Cr were associated with ↑ Aβ deposition on PET (flutemetamol) in amyloid-positive (on PET) cognitively unimpaired participants. ↑ mIns/Cr was associated with ↑ Aβ deposition on PET (flutemetamol) and in CSF-Aβ42-positive cognitively unimpaired participants.
[72]	CU (n = 16), aMCI (n = 11)	3T; TR/TE = 2000/32ms, single voxel, 2D-PRESS	Bilateral hippocampi	No difference in mIns/Cr between *APOE e*4 allele carriers and non-carriers
[73]	CU (n = 21); aMCI (n = 15)	3T, TR/TE= 3000/68 ms, single voxel, MEGA-PRESS	Posterior cingulate gyrus	↓ NAA was lower in Aβ-positive subjects compared to Aβ-negative (PiB PET) subjects. ↓ NAA was in *APOE e*4 allele carriers compared to non-carriers.
[74]	*APOE e*4 allele non-carriers (n = 89); *APOE e*4 allele carriers (n = 23)	3T, TR/TE= 1600/30 ms, single voxel, PRESS	Posterior cingulate gyrus	↑ Cho/Cr and ↑ mIns/Cr increase with age in *APOE e*4 allele carriers. ↑ Cho/Cr ratio *APOE e*4 carriers compared to non-carriers.
[29]	No to low likelihood of AD (n = 17); intermediate to high likelihood of AD (n = 24)	3T, TR/TE= 2000/30 ms, single voxel, PRESS	Posterior cingulate gyrus	↓ NAA/Cr and NAA/mIns were associated with ↓synaptic integrity and ↑higher p-tau pathology. ↑ Aβ burden was associated with ↑ mIns/Cr and ↓ NAA/mIns. ↑ GFAP-positive astrocytic burden showed a trend of association with decreased NAA/Cr and NAA/mIns.
[75]	CU (n = 17); AD (n = 19)	3T, TR/TE= 2000/30 ms, single voxel, PRESS	Hippocampus, posterior cingulate gyrus, and right parietal gyrus	↓ NAA/Cr (hippocampus) was correlated with ↓ CSF Aβ42. ↓ NAA/Cr (parietal gyrus) was correlated with ↑ CSF p-tau. ↑ mIns/Cr (posterior cingulate gyrus) was correlated with ↑ t-tau.
[76]	All subjects (n = 109); AD dementia (n = 40); non-AD dementia, (n = 14); MCI of AD type (n = 29) MCI of non-AD type (n = 26)	1.5T, TR/TE= 2000/272, single voxel, PRESS	Medial temporal lobe	↓ NAA was correlated with ↓ CSF Aβ42 in patient with AD dementia.
[77]	CU (n = 311)	1.5 T, 2000/30 ms, single voxel, PRESS	Posterior cingulate gyrus	↑ mIns/Cr and ↑ Cho/Cr were associated with ↑ Aβ load on PET (PIB).
[78]	Low AD likelihood (n = 11); intermediate AD likelihood (n = 9); high AD likelihood (n = 34)	1.5 T/2000/30 ms, single voxel, PRESS	Posterior cingulate gyrus	↓ NAA/Cr, ↑ mIns/Cr, and ↓ NAA/mIns in postmortem frequent neuritic plaque group compared to neuritic sparse plaque group. ↓ NAA/Cr in frequent neuritic plaque group compared to neuritic moderate plaque group. ↑ mIns/Cr and ↓ NAA/mIns in neuritic moderate plaque group compared to neuritic sparse plaque group. ↓ NAA/Cr, ↑ mIns/Cr, and ↓ NAA/mIns in high-likelihood AD group compared to low-likelihood AD group. ↑ mIns/Cr in high-likelihood AD group compared to intermediate-likelihood AD group. ↓ NAA/Cr, ↑ mI/Cr, and ↓ NAA/mI ratios were associated with higher Braak NFT stage, higher neuritic plaque score, and greater likelihood of AD.
[79]	CU (n = 61); patient group (MCI + AD dementia (n = 46)	1.5 T/2000/30 ms, single voxel, PRESS	Posterior cingulate/precuneus	No differences were noted on ^1^H-MRS metabolite ratios (NAA/Cr, mIns/Cr, NAA/mIns) across *APOE e*4 carriers and non-carriers.
[50]	CU (63); MCI (21); AD dementia (21)	1.5 T/2000/30 or 135 ms, single voxel, PRESS	Posterior cingulate gyrus; medial occipital; left superior temporal lobe	↑ NAA/Cr ratios (medial occipital) in patients with AD dementia correlated with *APOE e*4 carrier status.
[80]	Postmortem brain with AD pathology (49); non-demented control (5)	In vitro, 11.7 T, perchloric acid extracts	Autopsy brain samples from various brain regions	↑ mIns, ↑ GPC, and ↓ Glu in *APOE e*3/*e*3 samples from AD dementia patients compared to samples from normal control brain samples. ↓ NAA in *APOE e*3/*e*3 and *APOE e*4/*e*4 AD samples from AD dementia patients compared to samples from normal control brains (*APOE e*3/*e*3).

Abbreviations: Aβ: Amyloid-beta; AD: Alzheimer’s disease; aMCI: Amnestic Mild Cognitive Impairment; APOE: Apolipoprotein E; A−T−N−: Negative for amyloid-beta, tau, and neurodegeneration markers; A+T+N−: Positive for amyloid-beta and tau, negative for neurodegeneration markers; A+T+N+: Positive for amyloid-beta, tau, and neurodegeneration markers; CU: Cognitively unimpaired; CSF: Cerebrospinal fluid; FIDLOVS: Free Induction Decay Localized by Outer Volume Suppression; GFAP: Glial fibrillary acidic protein; GPC: Glycerophosphocholine; MCI: Mild cognitive impairment; MEGA-PRESS: Mescher–Garwood Point-Resolved Spectroscopy; MRSI: Magnetic resonance spectroscopy imaging; mIns: myo-inositol; NAA: N-Acetylaspartate; NfL: Neurofilament light; NFT: Neurofibrillary tangles; PET: Positron emission tomography; PiB: Pittsburgh compound B; PRESS: Point-Resolved Spectroscopy Single-Voxel Sequence; p-tau: Phosphorylated tau; RO948: Tau PET ligand; tCr: Total creatine; sLASER: adiabatic selective refocusing sequence; TR/TE: Repetition time/echo time; VOI: Volume of interest; MMSE: Mini Mental State Examination; MRS: Magnetic resonance spectroscopy; GABA: γ-Aminobutyric acid; Glx: Glu (glutamate) + Gln (glutamine); ↑: increase; ↓ decrease.

### 2.3. Cho

Choline is considered a biomarker for cell membrane (phospholipid) turnover, white matter integrity, and cellular density [53,81,82]. The peak at 3.2 ppm is assigned to mobile choline-containing compounds including PCho and GPC ascribed as total Cho (tCho), which are found in the myelin and the cell membrane [30,81,82,83]. While some studies reported a change in Cho signal related to AD pathology, the direction of change is not always consistent. Some reported an increase [47,50,84,85,86,87], and others reported a decrease or no change [36,39,41,43,88,89] in patients with AD dementia compared to controls. The elevation of Cho in AD dementia may be due to an increased membrane catabolism in response to an increased demand for acetylcholine synthesis, which leads to an increase in PCho and GPC [28,90,91].

### 2.4. Glu, Gln, Glx

The spectral peaks of Glu (at 2.35 ppm) and Gln (at 2.45 ppm), measured through conventional ^1^H MRS sequences, overlap at commonly used clinical MR field strengths (1.5 T and 3T) [56]. Therefore, these two peaks are generally assigned as Glx (Glu + Gln). Glu, a precursor of GABA, is an excitatory neurotransmitter and is mainly synthesized through the Glu-Gln cycle [92,93]. Earlier studies reported a decrease in glutamate or Glx in patients with AD dementia and MCI compared to cognitively unimpaired participants [44,69,73,94,95]. 

### 2.5. GABA

GABA is a primary inhibitory neurotransmitter in the brain. The differentiation of GABA from other overlapping peaks of Glx (at 2.35 ppm), NAA, Cr, and PCr (at 3.02 ppm) at lower field strengths (≤3T) is challenging. Spectral editing methods or the employment of two-dimensional spectroscopy protocols are needed to resolve overlapping signals [96]. Riese et al. reported that GABA levels were lower in patients with amnestic MCI compared to elderly controls [73]. In contrast, a study observed no significant change in GABA levels between normal elderly participants and those with AD dementia [97].

### 2.6. GSH

Glutathione (GSH) is considered as an important antioxidant in the brain [98]. A decrease in GSH levels in a variety of brain regions, including the hippocampus, frontal cortex, posterior cingulate cortex, and anterior cingulate cortex, has been demonstrated in patients with AD dementia as compared with age-matched cognitively unimpaired participants [99,100,101]. However, a recent meta-analysis reported that there was no change in GSH peroxidase and GSH reductase activities and GSH levels in human specimens [102].

### 2.7. Lactate

Lactate is synthetized by the lactate dehydrogenase enzyme. Compared to other metabolites, there are limited studies regarding lactate change in AD. Due to its fast and dynamic turnover, special advanced methods might be needed to detect lactate. It is generally observed using advanced editing sequences such as MEGA-PRESS or two-dimensional MRS methods such as J-PRESS [52,53]. A study reported elevated lactate in patients with AD dementia compared to control subjects using the J-PRESS spectrum [65] while others failed to report a change [103,104]. No age-dependent change in lactate level in APP/PS1 transgenic mice, which express a chimeric mouse/human amyloid precursor protein and a mutant human presenilin 1, was observed, while there was an age-dependent decrease in lactate in wild-type mice [105]. The disruption of the mitochondrial pathway which leads to impaired energy metabolism might be an underlying reason for elevated lactate levels. However, more research is needed to investigate lactate alterations in AD using advanced technologies, such as hyperpolarized ^13^C magnetic resonance spectroscopy, which significantly enhances the lactate signal and enables real-time tracking of metabolic changes [106,107].

### 2.8. Cr

The Cr and PCr systems play a role in energy metabolism for adenosine triphosphate regeneration and act as an energy buffer [108]. Since there are overlapping singlets of Cr and PCr peaks at 3.03 ppm and 3.9 ppm at 3T and lower fields, their sum ascribed as Cr or total Cr (tCr) is used in many studies, generally as an internal reference metabolite to calculate relative metabolite levels (metabolite/tCr) [27,109]. While tCr remains constant in various diseases such as AD dementia [38,39,41,109], it has been shown that total Cr levels may change with age [66,110,111,112] and white matter hyperintensity volume [113]. It is highly recommended that tCr levels should be used as an internal reference after confirming that its concentration (relative to water) is not changed [114].

## 3. Association of ^1^H MRS Metabolites with Postmortem Neuropathology 

Correlation studies between antemortem ^1^H MRS metabolite alterations and postmortem neuropathology are limited. Histopathological findings serve as a gold standard to validate ^1^H MRS findings to monitor the AD continuum and better understand how metabolite changes are associated with topographical neuropathological alterations [29,78].

We reported that a decrease in NAA/Cr and an increase in mIns/Cr (posteriorcingulate gyrus) correlates with postmortem Alzheimer-type pathology including the postmortem Braak neurofibrillary tangle stage, higher neuritic plaque score, and greater likelihood of AD (Figure 1) [78]. The study suggested that the mIns/Cr ratio may be more sensitive to early pathologic changes than the NAA/Cr ratio. Melissa et al. showed that antemortem ^1^H MRS metabolites (e.g., NAA/Cr and NAA/mIns) were linked to postmortem AD neuropathology, including the amyloid burden, synaptic integrity, and tau pathology [29]. In particular, the study identified a correlation between increased mIns/Cr and decreased NAA/mIns in the posterior cingulate gyrus with postmortem amyloid burden. Additionally, the study found an association between NAA/Cr and synaptic vesicle immunoreactivity but not neural density in the posterior cingulate gyrus across the entire cohort, including AD patients and control subjects. No such association was observed between Cho/Cr and mIns/Cr ratios and synaptic vesicle immunoreactivity which serves as a synaptic integrity marker. Furthermore, a higher postmortem pTau burden was associated with lower NAA/Cr and NAA/mIns ratios, while Cho/Cr was not associated with postmortem pTau. The study reported that postmortem late extracellular neurofibrillary tangle pathology was not correlated with NAA/Cr, mIns/Cr, Cho/Cr, and NAA/mIns. There was also no association between CD68 (a marker for activated phagocytic microglia)-positive microglia and any of the metabolite ratios studied in the study (i.e., NAA/Cr, mIns/Cr, Cho/Cr, and NAA/mIns) [29]. One of the important findings of this study was the association between NAA/Cr and a synaptic integrity marker and pTau but not neural loss or late extracellular neurofibrillary tangle pathology. Supporting evidence of these findings was reported later in an in vivo study using tau (Flortaucipir) PET [63].

## 4. Association of ^1^H MRS Metabolites with Tau and Amyloid PET 

### 4.1. NAA

In a cognitively unimpaired cohort, there was no statistically significant association between NAA/Cr (posterior cingulate gyrus) and Aβ deposition on PET imaging (11 C Pittsburgh compound B, henceforth PiB) after adjusting for sex and age [77]. In line with this study, it was reported that neither global cortical Aβ nor local (posterior cingulate gyrus) Aβ load on PiB PET were correlated with NAA (posterior cingulate gyrus) in a cohort consisting of cognitively unimpaired participants and those with amnestic MCI [73]. However, the study reported that NAA was lower in Aβ-positive participants compared to Aβ-negative participants. Zeydan et al. examined the ^1^H MRS metabolite profile in the posterior cingulate gyrus in two groups (i.e., cognitively unimpaired participants and participants with amnestic MCI) using an advanced sLASER MRS protocol [69]. The study reported that the level of NAA, mIns, Cr, and Cho between amnestic MCI participants, who were Aβ-positive on PiB PET, and cognitively unimpaired participants, who were Aβ-negative on PiB PET, were not statistically significant [69]. We recently reported that Aβ deposition on PET was not associated with NAA/tCr ratios in the posterior cingulate gyrus of cognitively unimpaired participants, while higher tau PET load was associated with a lower NAA/tCr ratio [63]. Extending these findings, a current study reported no significant association between tNAA/tCr with Aβ load on PIB PET within the gray matter (posterior cingulate gyrus or dorsolateral prefrontal cortex) [59]; however, this study reported a decrease in the tNAA/tCr ratio in patients with AD dementia in the gray matter. These studies suggest that an increase in Aβ load may not be directly associated with NAA in the preclinical and prodromal stages of AD pathology.

Current studies have investigated the association between ^1^H MRS and both tau and Aβ loads on PET in cognitively unimpaired individuals. Our group recently investigated the association between brain metabolites with tau and Aβ load on PET [63]. An increase in the posterior cingulate gyrus tau load on Flortaucipir PET was associated with lower NAA/tCr in cognitively unimpaired older adults [63] (Figure 2). Extending these findings, a decreased NAA/Cr ratio in the posterior cingulate gyrus was associated with elevated tau and Aβ load on PET in a cohort consisting of participants with non-AD and AD dementia who were categorized based on their A/T/N status based on PET and MRI [62]. The study reported that the NAA/Cr ratio in early AD (A+T+N−) and late AD (A+T+N+) was lower compared to controls (A−T−N− and A+T−N−). An association between elevated NAA/Cr and an increase in global Aβ load on PET and tau load on PET was present in the whole cohort. Furthermore, the study reported that the NAA/Cr ratio could be used to discriminate A−T−N− and A+T−N− from participants with early AD (A+T+N−).

### 4.2. mIns

In a cognitively unimpaired cohort, elevated mIns/Cr in the posterior cingulate gyrus was associated with an increased Aβ load on PiB PET [77]. Voevodska et al. reported an association between higher Aβ load on PET with higher mIns/Cr and mIns/NAA (posterior cingulate gyrus) ratios in cognitively unimpaired participants who were classified as Aβ positive based on Aβ CSF levels [71]. The study also reported that when Aβ positivity was based on PET with Flutemetamol, the association between elevated mIns/Cr and increased Aβ load in cognitively unimpaired controls was sustained [71]. However, these associations were not present in the CSF-Aβ42-negative cognitively unimpaired controls, suggesting that a certain level of neuropathological accumulation driven by amyloid load may be required to observe these associations [71]. Nedelska et al. reported that elevated mIns/Cr and lower NAA/mIns (posterior cingulate gyrus) in cognitively unimpaired participants at baseline were associated with baseline Aβ load and an increased rate of Aβ deposition on PiB PET over time [70]. Extending these findings, a higher level of mIns/Cr (posterior cingulate gyrus) was reported in participants with biologically defined late AD dementia (A+ T+ N+) with cognitive impairment compared with cognitively unimpaired participants (A−T−N− and A+T−N−) [62], but no statistically significant difference in mIns/Cr ratio was observed between cognitively unimpaired and biologically defined early AD participants (A+T+N−). The study also demonstrated an association between higher mIns/Cr and higher global Aβ load on PET (PiB) and tau load on PET in the entire cohort.

### 4.3. Cho

The Cho/Cr ratio in the posterior cingulate gyrus was associated with an increased Aβ load on PET (PiB) in a relatively large cognitively unimpaired cohort (n = 311) [77]. Partially in line with this study, Voevodskaya et al. reported that elevated Cho/Cr in the posterior cingulate gyrus region in cognitively unimpaired participants was associated with an increased Aβ load on PET with flutemetamol but only in amyloid-PET-positive cognitively unimpaired participants [71]. Interestingly, when the CU group was classified as amyloid-positive based on CSF Aβ42 levels instead of PET, Cho/Cr was no longer associated with Aβ load on PET or CSF Aβ load [71]. This suggests that Aβ load in CSF and PET might be capturing distinct aspects of amyloid pathology. In addition, the variability in CSF Aβ42 level in these participants might have influenced the relationship between Cho/Cr and amyloid load, making the association less detectable in the Aβ-positive group based on CSF analysis. 

Spotorno et al. reported no correlation between tCho/tCr and Aβ and tau load on PET and no moderation effect of *APOE ε*4 genotype on these associations in a cohort consisting of CU (Aβ negative and Aβ positive on PET) and MCI (Aβ positive on PET) [60]. Most recently, Chen et al. reported no change in Cho/Cr between normal (A−T−N− and A+T-N-), biologically defined early AD (A+T+N−), and late AD (A+T+N+) groups [62]. The results of Sportorno et al. are not fully in line with those of Kantarci et al. (2011) which might be attributed to differences in the characteristics of participants among the studies. The participants in the study by Kantarci et al. (2011) were relatively older than those enrolled in the study by Sportorno et al. (2022). This suggests that the association between Cho and Aβ load on PET might be more detectable when neuropathological alterations have progressed further in older participants. 

### 4.4. Glx and Glu

Rieze et al. reported no association between Glx and Aβ deposition on PiB PET (global and local [posterior cingulate gyrus]) in a cohort consisting of participants with amnestic MCI and cognitively unimpaired participants [73]. Zeydan et al. reported a decrease in the Glu and Glu/mIns ratio in the amnestic MCI group (Aβ positive on PET with PiB) compared to the cognitively unimpaired group (Aβ negative on PET with PiB) in the posterior cingulate gyrus [69]. A decrease in the Glu/mIns ratio was associated with a higher global cortical Aβ deposition in the whole cohort consisting of amnestic MCI and cognitively unimpaired participants [69]. In addition, the study reported that this correlation was not present when groups (participants with amnestic MCI and cognitively unimpaired participants) were analyzed independently. Only a few studies have investigated the association between ^1^H MRS metabolites and both tau and amyloid loads on PET in cognitively unimpaired individuals and patients with AD. An increase in the posterior cingulate gyrus tau deposition on PET with 18F-flortaucipir was associated with lower Glu/tCr ratios in cognitively unimpaired older adults [63] (Figure 2), and biological sex modified this association. However, the association between Glu/tCr and Aβ deposition on PET with PIB was not statistically significant. Chen et al. categorized their cohort as cognitively unimpaired controls (A−T−N− and A+T−N−), early AD (A+T+N−), and late AD (A+T+N+) dementia using PET and MRI data. The study reported no difference in Glu/tCr across groups (controls, biologically defined early AD and late AD). Matsuaoka et al. reported that a decrease in Glu/tCr in the posterior cingulate cortex was associated with an increase in tau load on PET with florzolatau in participants with AD dementia [59]. Riese et al. (2015) studied a cohort of MCI and healthy controls and found no difference in Glx between groups categorized as Aβ negative and Aβ positive on PET [73].

### 4.5. GABA

GABAergic dysfunction has been reported in the AD continuum [115]. Some studies reported lower GABA/Cr in patients with AD dementia [115], while others found no change in GABA levels compared to controls [97]. Rieze et al. reported similar GABA levels in the posterior cingulate gyrus among groups classified as Aβ positive and negative on PET [73]. The study also reported no correlation between GABA and Aβ deposition on PiB PET (global and local [posterior cingulate gyrus]) in a cohort consisting of participants with amnestic MCI and no cognitive impairment. 

### 4.6. GSH

In cognitively unimpaired participants, a negative correlation between GSH levels (posterior cingulate gyrus) and brain amyloid load on PET (PiB) in the temporal and parietal regions was reported, suggesting preclinical changes in GSH level might be an early biomarker of AD pathology [68]. In a recent study, no difference in GSH/tCr (posterior cingulate gyrus) across groups (controls: A−T−N− and A+T−N−; biologically defined early AD: A+T+N− and late AD: A+T+N+) was found [62]. Further research is needed to explore the relationship between GSH levels and Aβ and tau pathology [102]. 

## 5. Association of ^1^H MRS Metabolites with Biofluid Biomarkers

The emergence of blood-based plasma biomarkers represents a major recent breakthrough in identifying biological indicators of AD. These biofluid biomarkers are non-invasive, readily accessible, and cost-effective, making them crucial for detecting AD throughout its preclinical, prodromal, and dementia stages. The most extensively evaluated AD-related plasma biomarkers include Aβ, especially Aβ_40_, Aβ_42,_ and their ratio Aβ_42_/Aβ_40_, and phosphorylated tau (p-tau) protein at epitopes 181, 217, and 231 (p-tau181, p-tau217, and p-tau231) which reflect neuritic plaques and neurofibrillary tangle pathologies [116,117,118]. It has been demonstrated that the Aβ_42_/Aβ_40_ ratio in plasma correlates with CSF AD biomarkers and amyloid PET [119,120,121]. Similarly, the level of p-tau proteoforms was associated with CSF, PET, and postmortem AD neuropathological markers [122,123,124,125].

The plasma neurofilament light chain (NfL) and glial fibrillary acidic protein (GFAP) have been commonly studied in AD research [116,117,118]. The NfL is a marker of neuroaxonal damage, and elevated NfL is associated with the progression of neurodegenerative disorders such as AD, Huntington’s disease, and multiple sclerosis [126,127]. Interestingly, it has been reported that plasma NfL but not CSF NfL was significantly associated with cognition [128], suggesting CSF and plasma biomarkers might provide complementary information rather than being directly interchangeable. Plasma GFAP, a marker of astrocytic activation, is associated with the elevated risk and severity of AD-type and non-AD-type dementia [129,130]. While biofluid AD markers in plasma can also be measured in CSF, lumbar punctures, however, are burdensome and costly and require specialized training, limiting their use if serial assessment is required.

### 5.1. NAA

Lower medial temporal lobe NAA was correlated with lower CSF Aβ42 within patients with AD dementia [76]. Neither CSF tau nor CSF pTau181 were correlated with NAA within all dementia groups (i.e., AD dementia, non-AD dementia, MCI of AD type, and MCI of non-AD type) or any individual dementia groups. Bittner et al. studied the correlation of hippocampal, posterior cingulate gyrus, and right parietal gyrus NAA with CSF Aβ42 and CSF p- and t-tau in cognitively unimpaired and AD dementia patients [75]. A lower hippocampal NAA/Cr ratio in patients with AD dementia was associated with lower CSF Aβ42 levels but not with CSF p-tau or t-tau, whereas lower parietal NAA/Cr was associated with higher CSF p-tau but not with CSF Aβ42 or t-tau [75]. Voevodska et al. (2016) showed that the NAA/Cr ratio in the posterior cingulate gyrus was lower in MCI (CSF Aβ42 positive) compared to cognitively unimpaired individuals (CSF Aβ42 negative) [71]. Voevodskaya et al. (2019) reported that the estimated rate of change in NAA/Cr in the posterior cingulate gyrus was −2.0%/year in a cohort consisting of cognitively normal controls, mild cognitive impairment, and subjects with cognitive decline who were classified based on their CSF Aβ load as CSF Aβ positive or CSF Aβ negative at baseline [64]. However, the estimated rate of change of NAA/Cr was not significant in the CSF-Aβ-negative group. The metabolite ratios were also compared within CSF-Aβ-positive and -negative groups at baseline (visit 1), visit 2, and visit 3 with a gap of approximately 2 years between visits. It was shown that the NAA/Cr ratio was lower in the CSF-Aβ-negative group compared to the CSF-Aβ-positive group at visit 2 and visit 3 [64]. 

Hone-Blanchet reported that tNAA in the medial frontal cortex did not change among older women categorized as cognitively unimpaired Aβ positive, cognitively unimpaired Aβ negative, and MCI (Aβ positive) groups [61]. Furthermore, CSF Aβ 42 levels were not associated with the level of tNAA and other metabolites (e.g., tCho, tNAA/mIns, Glx, mIns, GABA, GABA/tCr) [61]. Matsuoka et al. showed that there was a significant association between increased plasma NfL and a decreased tNAA/tCr in the right dorsolateral prefrontal cortex of participants with AD dementia; however, there was only a trend of association between elevated plasma NfL and decreased tNAA/tCr from the posterior cingulate gyrus, but this correlation did not reach statistical significance.

These studies highlight the value of integrating data from AD-specific CSF and plasma markers. A combination of both fluid and MRS biomarkers can track the progression of cognitive decline during the preclinical and prodromal phases of AD [64]. These studies also show that there are regional differences regarding how MRS metabolites correlated with AD CSF biomarkers might be related to the regional progression of NFT and amyloid pathology. 

### 5.2. mIns 

A serial MRI/MRS study was conducted in cognitively unimpaired individuals for 7 years [67]. Seven years after the baseline measurements, CSF and ^1^H MRS data were collected in subjects who were converted to MCI/AD, Parkinson’s disease, and dementia with Lewy bodies. The study demonstrated that CSF Aβ42 and CSF p-tau were not correlated with the NAA/mIns ratio in this cohort. Voevodska et al. (2019) showed that there were no differences between NAA/mIns in the posterior cingulate gyrus/precuneus region between CSF-Aβ42-positive and Aβ42-negative participants (60 years or older) at baseline (visit 1) in a longitudinal design [64]. However, approximately 2.3 years (visit 2) and approximately 4 years (visit 3) after the baseline visit, a decrease in NAA/mIns in CSF-Aβ42-positive compared to CSF-Aβ42-negative participants was observed. The study reported that being CSF Aβ42 positive at visit 1 was associated with a decrease in NAA/mIns over time in all cohorts (CSF-Aβ42-positive and -negative cases) (the model was adjusted for baseline age, sex, and *APOE ε*4 carriership). Furthermore, the study reported a higher rate of change in the MCI CSF-Aβ42-positive participants compared to MCI CSF-Aβ42-negative participants [64]. Hone-Blanchet reported that the mIns/tCr ratio in the medial frontal cortex was elevated in MCI (CSF Aβ positive) compared to cognitively unimpaired CSF-Aβ-negative and Aβ-positive women [61]. In another study, an increase in plasma GFAP associated with elevated mIns/tCr in the posterior cingulate gyrus/precuneus region in a cohort consisting of cognitively unimpaired (Aβ negative and positive on PET with flutemetamol) and MCI (Aβ positive on PET) participants who were *APOE ε*4 carriers [60].

### 5.3. Cho

Voevodska et al. (2016) reported that Cho/Cr in participants with MCI (all CSF Aβ42 positive) was higher compared to cognitively unimpaired participants who were CSF Aβ42 negative [71]. Voevodska et al. (2019) showed that there were no differences between Cho/Cr in the posterior cingulate gyrus/precuneus region between CSF-Aβ42-positive and CSF-Aβ42-negative participants (60 years or older participants) at baseline (visit 1) [64]. However, approximately 2.3 years (visit 2) after the baseline visit, an increase in the Cho/tCr ratio in CSF-Aβ42-positive compared to CSF-Aβ42-negative participants was observed when the groups were compared with each other at the same visit. There was no difference in Cho/Cr ratios among CSF-Aβ42-positive and -negative groups at visit 3 (approximately 4 years after the baseline) [64].

### 5.4. Glu

Matsuoka et al. showed that there was a significant association between increased plasma NfL and a decreased Glu/tCr in the right dorsolateral prefrontal cortex of participants with AD dementia, and there was only a trend of association between elevated plasma NfL and decreased Glu/tCr in the posterior cingulate gyrus, but this correlation did not reach statistical significance [59].

### 5.5. GABA

Hone-Blanchet et al. demonstrated that GABA levels in the medial frontal cortex of participants with MCI were lower compared to cognitively unimpaired participants (CSF Aβ42 positive and Aβ42 negative) [61]. While older age was correlated with lower GABA levels in both cognitively unimpaired CSF-Aβ42-positive and -Aβ42-negative participants, CSF biomarkers (Aβ42, t-tau and p-tau) were not associated with GABA and GABA/Cr levels.

## 6. Influence of *APOE ε*4 Allele on ^1^H MRS Metabolites

Carrying one or two copies of the *APOE ε*4 allele elevates the risk factor for late-onset AD dementia. A recent study showed that almost all participants who were homozygotes for the *ε*4 allele (*ε*4/*ε*4) exhibited AD pathology (A+T+N+) [58]. However, only a few studies investigated whether *APOE ε*4 carrier status affects the metabolite levels or *ε*4 carrier status modifies the relationship between ^1^H MRS metabolites and AD biomarkers. Some studies reported no effect of *APOE ε*4 carrier status on metabolite levels, and their association with AD biomarkers (Aβ and tau load), but others reported that *APOE ε*4 allele carrier status affects the metabolite levels and/or the relationship between the metabolites and AD biomarkers [50,58,60,66,67,70,71,72,73,79,80].

No differences in metabolite ratios (NAA/Cr, mIns/Cr, NAA/mIns) were found across the *APOE* genotype (i.e., *ε*4 carriers and non-carriers) within cognitively unimpaired control and patient (MCI+AD dementia) groups [79]. Riese et al. reported no difference in GABA and Glx levels between *APOE ε*4 carriers and non-carriers in a cohort consisting of cognitively unimpaired individuals and those with amnestic MCI [73]. In another study, no difference in mIns/Cr between *APOE ε*4 carriers and non-carriers was found in a cohort that included both cognitively unimpaired participants and subjects with amnestic MCI [72]. In line with this study, Voevodska et al. reported that *APOE ε*4 allele carrier status did not affect the mIns/Cr levels (posterior cingulate gyrus) across cognitively unimpaired CSF-Aβ42-positive and CSF-Aβ42-negative MCI groups (CSF Aβ42 positive) [71]. Nedeslska et al. reported that *APOE ε*4 allele carrier status did not modify the relationship between MRS metabolites (NAA/mIns, mIns/Cr) and the rate of Aβ deposition on serial PET [70]. A serial MRI/MRS study was conducted in cognitively unimpaired individuals for 7 years [67]. At baseline, the mIns/Cr ratio was elevated in subjects with two copies of the *APOE ε*4 allele compared to non-carriers. Additionally, the NAA/mIns ratio was significantly decreased in subjects who were homozygous for the *APOE ε*4 allele compared to those who were heterozygous for the *APOE ε*4 allele and non-carriers. However, the NAA/Cr ratio showed no significant difference between subjects with and without the *APOE ε*4 allele [67]. Suri et al. showed that there was no significant effect of three *APOE* groups (*e*3 carrier, *ε*3 homozygotes, *ε*4 carriers) or interaction between *APOE* groups and age on the metabolite profile in the posterior cingulate gyrus in individuals who were younger (between 20 and 40 years old) and cognitively unimpaired older age cohort (between 60 and 85 years old) [66]. In a cohort composed of subjects without cognitive impairment and with MCI who were *APOE*
*ε*4 allele carriers, no association between tau load on PET with mIns/tCr (posterior cingulate gyrus) was observed [60]. 

In postmortem perchloric acid brain extracts, an increase in mIns and GPC and a decrease in Glu and NAA was observed in AD brains with *APOE e*3/*e*3 allele carriers status compared to normal control brains with *APOE e*3/*e*3 allele carrier status [80]. The study also reported differences between *e*3/*e*3 AD and *ε*4/*ε*4 AD brains. For example, NAA was lower, and GPC was higher in *e*4/*e*4 AD brains compared to *ε*3/*ε*3 AD brains. We reported that the NAA/Cr ratio of patients with AD dementia significantly correlated with *APOE ε*4 carrier status [50]. Riese et al. reported that NAA levels were lower in a cohort of participants who were cognitively unimpaired and those with amnestic MCI, who had the *APOE ε*4 allele compared to those without it [73]. A recent study compared the metabolite ratios of cognitively unimpaired groups who were carrying two copies of the *APOE ε*4 allele (i.e., *APOE ε*4 homozygotes) with non-carriers [67]. The study reported a higher mIns/Cr in *APOE ε*4/*ε*4 homozygotes compared to non-ε4 carriers. Furthermore, a decrease in the NAA/mIns ratio was reported in those with *ε*4/*ε*4 carriers compared with subjects with only one copy of the *ε*4 allele. A recent study using voxel-wise analysis demonstrated an association between elevated Aβ load on PET with an increased mIns/tCr ratio (posterior cingulate gyrus) only in the *APOE ε*4 allele carrier group (cognitively unimpaired + MCI) [60]. A recent study investigated the influence of *APOE ε*4 carrier status on the relationship between GABA and Glu (posterior cingulate gyrus) and Aβ load on PET. The study reported that elevated gray matter GABA and Glu were associated with higher Aβ load on PET with positive effect modification by *APOE ε*4 *allele* carrier status [58].

More research is needed to understand the impact of *APOE ε*4 on ^1^H MRS metabolites. While some findings suggest significant alterations in certain metabolite ratios among *APOE ε*4 carriers, particularly homozygotes, further research is needed to clarify these relationships and their implications for understanding and diagnosing AD.

In summary, the studies reviewed above highlight several key findings in ^1^H MRS research on the AD continuum. The most common findings of ^1^H MRS studies in the AD continuum are an increase in mIns/tCr in the medial temporal lobe and posterior cingulate cortex [27,131]. Elevated mIns/tCr precedes the decrease in NAA [50]. The current meta-analysis demonstrated that during progression from MCI to AD, NAA (or NAA/tCr) decreases in the hippocampus and posterior cingulate gyrus, while mIns (or mIns/tCr) increases in the posterior cingulate gyrus. Association studies have demonstrated a positive correlation between mIns/tCr and amyloid load and a negative correlation between synaptic metabolites such as NAA and Glu and tau load in cognitively unimpaired participants [63,77]. Studies associating premortem MRS and postmortem pathology have shown that the N-acetylaspartate–to–myo-inositol ratio is a strong predictor of the pathologic likelihood of AD [78]. Overall, these studies suggest that NAA, mIns, and Glu (or Glx) might be used as potential biomarkers to observe changes at the preclinical stage of AD.

Regarding biofluid biomarkers, our review also revealed several important findings. In summary, key ^1^H MRS metabolites show distinct regional associations with biofluid biomarkers in AD. Lower NAA/Cr in the hippocampus and posterior cingulate gyrus correlates with reduced CSF Aβ42 but shows no association with CSF tau or CSF pTau181 [75,76]. NAA/mIns decreases over time in CSF-Aβ-positive individuals, particularly in the posterior cingulate [67]. Elevated Cho/Cr is observed in CSF-Aβ-positive MCI [71]. Increased mIns/tCr is linked to higher plasma GFAP levels in APOE *ε*4 carriers [60]. Decreased Glu/tCr correlates with elevated plasma NfL in AD [59], but CSF biomarkers (Aβ42, t-tau, and p-tau) were not associated with GABA and GABA/Cr levels [61]. These findings reflect the regional variability and complexity of metabolite changes in AD.

## 7. Future Directions

In recent years, substantial progress has been made in the identification of AD biomarkers, including PET, CSF, and plasma biomarkers, alongside growing evidence from ¹H MRS studies. These advancements highlight the potential for integrating ¹H MRS findings with Aβ and tau pathology biomarkers, providing a more comprehensive understanding of AD progression across different stages and regions of the brain. Plasma biomarker testing is relatively easy to scale for large populations, as it involves basic blood collection and standard laboratory analysis techniques. Research is ongoing to test plasma biomarkers’ sensitivity and specificity for longitudinal observations. Both plasma biomarkers and MRS biomarkers provide complimentary information but in different forms. ¹H MRS uses standard MRI machines, which are already widely available in most clinical settings. While plasma biomarkers are relatively inexpensive and easy to scale, they do not provide localized information about brain metabolite alterations, which is critical for understanding region-specific pathologies in diseases like AD. Well-established MRS protocols allow longitudinal observations within the same cohort. The additional software for metabolite analysis is relatively inexpensive compared to PET imaging, which requires expensive radioisotopes and specialized equipment. In terms of potential clinical income, ^1^H MRS can be integrated into routine MRI scans, offering a cost-effective option for longitudinal studies and clinical trials without significant additional operational costs. 

AD biomarker (PET, CSF, plasma) and MRS studies suggest that the association of ^1^H MRS with Aβ and/or tau pathology may vary based on the AD stage and the topographical heterogeneity of the disease, with these associations being region-specific. For example, some studies observed an association between ^1^H MRS metabolites, such as NAA, and Aβ load on PET in a cohort involving participants with the prodromal and AD dementia stages of the disease. In contrast, the association between ^1^H MRS metabolites (NAA/Cr and Glu/Cr) and tau load on PET was detected even at the preclinical stage in cognitively unimpaired participants. These findings indicate that correlations between AD biomarkers and ^1^H MRS metabolites vary by region, which may be related to the spatial progressions of amyloid and tau pathologies. For example, amyloid pathology progresses from the neocortical regions to the limbic and subcortical regions, while tau pathology begins in the transentorhinal cortex and spreads to the paralimbic and neocortical areas [132,133]. While considering the association between metabolic changes in the various brain regions with AD biomarkers (PET, CSF, plasma), it is crucial to consider the spatial and temporal dynamics of amyloid and tau pathologies to understand the underlying mechanisms of AD progression.

Various studies reported conflicting findings regarding the association of brain metabolites with Aβ/tau pathology. These differences can be partially attributed to variations in cohort characteristics, acquisition and quantification methodologies, disease stage and progression, and genetic factors. Variability in the studied populations’ characteristics, including differences in age, cognitive status (cognitively unimpaired, MCI, AD dementia), genetic factors (e.g., *APOE* status and carrying one of two copies of the *ε*4 allele), and race, can influence the outcomes. Some studies focused on cognitively unimpaired individuals, while others included participants with amnestic MCI or MCI and AD dementia or mixed groups. There are also methodological variations between studies. Some studies used higher-field MRS, and others used relatively lower-field MRS with varying acquisition protocols and regions of interest. Advanced techniques such as the sLASER protocol coupled with the automated volume of interest prescription may provide increased sensitivity and specificity compared to other methods [134]. Variations in the disease staging might be a source of contrasting findings. The relationship between metabolites and AD biomarkers may not be as pronounced as in later stages, where significant neuronal loss and metabolic changes are more evident. Understanding these differences may help in interpreting the results and drawing more comprehensive conclusions about the underlying biochemical processes in AD.

Ultra-high-field MR clinical systems (7T and higher) offer promising opportunities, including enhanced spectral resolution, improved signal-to-noise ratio, and reliable quantification of low-concentration metabolites like Glu, Glc, Gln, GSH, and GABA. Further research is required to fully understand and harness the clinical potential of ultra-high-field MRS in the AD continuum.

¹H MRS holds significant promise for monitoring disease progression, especially in clinical trials targeting early predementia pathology. Future studies should focus on evaluating the potential of ¹H MRS alongside plasma biomarkers in this setting, with careful consideration given to underrepresented racial and ethnic groups, as well as the role of biological sex as a variable [135].

There are ongoing efforts to harmonize, standardize, and optimize ^1^H MRS methods for both single-center and multicenter studies [136,137,138,139]. Future MRS studies should consider the consensus recommendations from experts to facilitate multicenter studies and ensure the reproducibility of results [137]. Recent advancements in ^1^H MRS including the automated volume of interest prescription pipeline [134], which enables fast and automated voxel placement, eliminates the requirement of manual voxel placement, and enables higher inter- and intra-subject consistency of voxel placement, would enhance the clinical integration of MRS and its use in clinical trials as an outcome measure [134]. Furthermore, future MRS studies can incorporate advanced MRS protocols such as modified sLASER to overcome the limitations of conventional MRS sequences such as chemical shift displacement errors at 3T and 7T [137].

Overall, integrating advancements in ^1^H MRS with recent developments in the AD biomarker field offers a comprehensive approach to understand disease progression and evaluate treatment strategies.

## Figures and Tables

**Figure 1 ijms-25-10064-f001:**
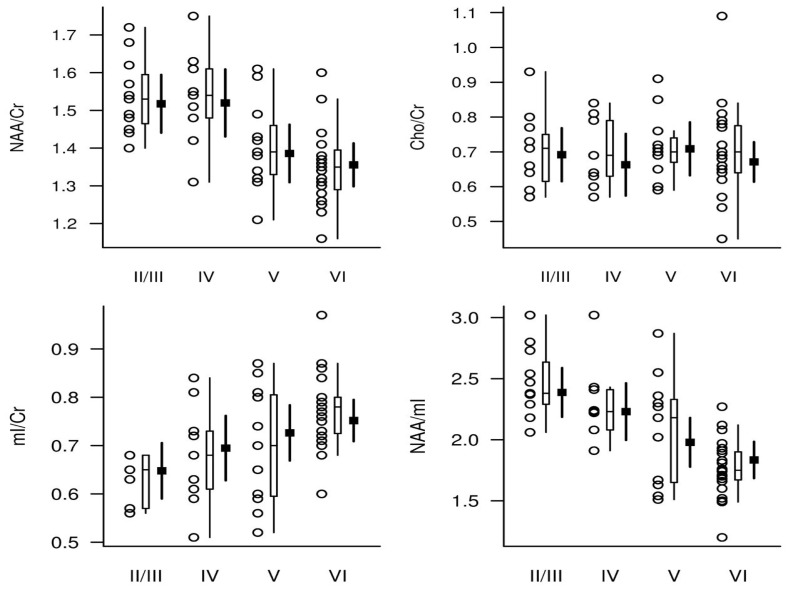
^1^H MR spectroscopic metabolite ratios plotted according to Braak NFT stage (horizontal axis). For each Braak NFT stage diagnosis, individual values (circles), a box plot of the distribution, and the estimated mean and 95% CI (darker lines) for the mean are shown. The mean and CI were derived from ANCOVA models and are assumed for a 78-year-old woman in whom the interval from ^1^H MR spectroscopy to death is 2 years. With permission from Radiology [78].

**Figure 2 ijms-25-10064-f002:**
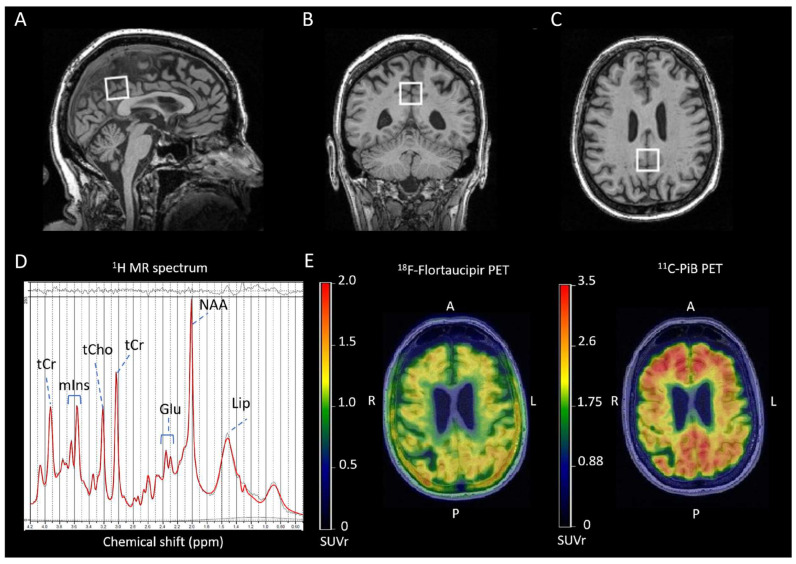
Sagittal (**A**), coronal (**B**), and transverse (**C**) T1-weighted magnetic resonance images with superimposed posterior cingulate gyrus magnetic resonance spectroscopy volume of interest (20 × 20 × 20 mm^3^) (white square). Representative 1H MRS spectra, AV-1451-PET, and PiB-PET of a clinically normal individual. The thick red curve on the representative MRS spectra is the LCModel fit to the data. The thin curve under the spectra is the baseline. The residual (data minus the fit to the data) is shown on the top of the spectra. (**D**) Single-voxel proton (^1^H) magnetic resonance (MR) spectrum acquired from the posterior cingulate gyrus of a cognitively unimpaired participant (age = 81) at 3 T with sLASER sequence. The thick red curve on the representative MR spectrum is the LCModel fit to the data. The thin curve under the spectrum is the fitted baseline. The residual (data minus the fit to the data) is shown at the top of the spectrum. The chemical shift axis is labeled in parts per million (ppm) unit. The Y axis is an intensity scale of each spectral line with no unit. (**E**) The representative cortical flortaucipir PET and PiB PET scans were acquired from the same participant. The participant had low NAA/tCr = 1.20 and Glu/tCr = 0.99, a high PCG flortaucipir standard value uptake unit ratio of 1.22, and a high PCG PiB standard value uptake unit ratio of 3.06. The PET scans were registered to the T1-weighted MR image and displayed together. We observed flortaucipir uptake in the skull of this participant. The meningeal and bone uptake of flortaucipir is a known manifestation of off-target binding. The cause is unknown. In contrast, such off-target uptake in PiB is not seen except with rare cases of bone uptake in diseases with high rates of bone remodeling (e.g., hyperostosis frontalis interna). The representative color scale shows the standardized uptake value ratios. Abbreviations: Glu, glutamate; Lip, lipid signal; NAA, N-acetylaspartate; PET, positron emission tomography; PiB, Pittsburgh compound-B; tCr, phosphocreatine + creatine; tCho, phosphocholine + glycerophosphocholine; mIns, myo-inositol; A, anterior;P, posterior; L, left; R, right. (For the interpretation of the references to color in this figure legend, the reader is referred to the Web version of this article.) This figure, originally appearing as Figure 1 and Figure 2, is reprinted from “1H MR spectroscopy biomarkers of neuronal and synaptic function are associated with tau deposition in cognitively unimpaired older adults”, Neurobiology of Aging, Volume 112, April 2022, Pages 16–26, with permission from Elsevier [63].

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
