# Peer review of "Understanding Proton Magnetic Resonance Spectroscopy Neurochemical Changes Using Alzheimer’s Disease Biofluid, PET, Postmortem Pathology Biomarkers, and APOE Genotype"

_ijms, 2024, doi:10.3390/ijms251810064_

Round 1

Reviewer 1 Report

Comments and Suggestions for Authors

The manuscript provides a comprehensive review of the research and application of Proton Magnetic Resonance Spectroscopy in the field of AD, and I would recommend the author to address the following issues when revising the manuscript:

1. I hope the author can make a summary of 1H MRS metabolites mentioned in the manuscript. Which metabolites can best reflect the early stage or preclinical stage of AD? Which metabolites have obtained more consistent results or conclusions from different studies, and are most promising to be translated into clinical applications?

2. At present, important progress has been made in the research of blood diagnostic markers for AD. Are there any advantages in terms of cost and income in the method of testing 1H MRS metabolites?

3. “Unlike biofluid biomarkers, which reflect the global pathology, 1H MRS data can provide regionally specific biochemical changes in the brain. How to understand" which reflect the global pathology ", such a statement seems inappropriate.

4. Please check the sentence "Furthermore, the study reported that NAA/Cr ratio". Is there a period missing here, or is the sentence not finished.

Author Response

Response to Reviewers Comments

1. Summary

2. Point-by-point response to Comments and Suggestions for Authors

Reviewer 1

 Comment 1:

I hope the author can make a summary of 1H MRS metabolites mentioned in the manuscript. Which metabolites can best reflect the early stage or preclinical stage of AD? Which metabolites have obtained more consistent results or conclusions from different studies, and are most promising to be translated into clinical applications?

Response 1:

Thank you for pointing this out. We have provided a summary of the relevant 1H MRS metabolites in the revised manuscript. These changes can be found on page [16], paragraph [3-4].

Comment 2:

At present, important progress has been made in the research of blood diagnostic markers for AD. Are there any advantages in terms of cost and income in the method of testing 1H MRS metabolites?

Response 2:

Thank you for the comment. Both techniques are cost effective compared to PET. ¹H MRS has several advantages in terms of cost and clinical utility. First, it uses standard MRI machines, which are already available in most clinical settings, reducing the need for expensive new infrastructure. In contrast, imaging methods like PET require specialized equipment and expensive radioisotopes, which significantly increase the cost per patient. While plasma biomarkers are relatively inexpensive and easy to scale, they do not provide information about brain metabolite alterations, which is critical for understanding region-specific pathologies in diseases like AD.

In terms of potential clinical income, ¹H MRS can be integrated into routine MRI scans, offering a cost-effective option for longitudinal studies and clinical trials without significant additional operational costs. The ability to monitor disease progression and therapeutic response through non-invasive brain metabolite tracking makes ¹H MRS an attractive tool for pharmaceutical companies and healthcare providers. Additionally, advancements in automated MRS protocols, such as automated voxel placement, further reduce the time and labor needed to acquire and analyze data, potentially lowering costs and increasing the feasibility of widespread clinical use.

We extended our “Future direction” section to point out advantages of MRS in terms of cost and income. The revised section can be found on page [16], paragraph [5].

Comment 3:

“Unlike biofluid biomarkers, which reflect the global pathology, xxbiochemical changes in the brain. How to understand 'which reflect the global pathology', such a statement seems inappropriate.”

Response 3:

Thank you for your comment. We agree that the statement could be clearer. The phrase has been rewritten to clarify that biofluid biomarkers provide systemic-level insights of changes in AD biomarkers, while 1H MRS allows for regionally specific biochemical assessments in the brain.

This revision is located on page [2], last sentence of the paragraph [3].

Comment 4:

Please check the sentence "Furthermore, the study reported that NAA/Cr ratio". Is there a period missing here, or is the sentence not finished?

Response 4:

Thank you for noticing this. A period was indeed missing. When the journal converted the text into the journal style, the missing part of the sentence merged to figure 2 legends. The sentence has been corrected.

Reviewer 2:

Comment 1:

Alzheimer’s disease (…) accounting for 60-80 of all dementia cases. More than 55 million people live with AD and other dementias (…)” (page 1). First, the authors should add the percentage symbol after “60-80”. Then, could the authors provide quantitative data insights about the worldwide incidence of disability-adjusted life years (DALYs) according to this disease? This will greatly aid the potential readers to better understand the significance of this work.

Response 1:

Thank you for this suggestion. We have added the percentage symbol and included the requested DALYs data to provide a more comprehensive understanding of AD’s global impact. The revised text can be found on page [1], paragraph [1].

Comment 2:

“The accumulation of amyloid-beta (…) neurofibrillary tangles (…) symptoms of AD dementia manifest” (page 2). Here, the authors are mainly focused on the action of Aβ fibrils and aggregates in the progression of Alzheimer’s disease, but there exist other misfolded proteins that can play this role, such as α-syn [1] or S100A9 [2] under the effect of calcium ions. [1] https://doi.org/10.1016/j.bbrc.2022.02.097

[2] https://doi.org/10.3390/biom14091091

Response 2:

Thank you for the insightful comment. We agree with the reviewer that AD patients at all stages of AD pathogenesis demonstrate clinical and pathological heterogeneity. The manuscript has been updated to include a brief discussion on additional misfolded proteins, such as α-syn and S100A9. These changes can be found on page [2], paragraph [3].

Comment 3:

“ Commonly studied 1H MRS metabolites in AD” (pages 2-8). Here, it should be also discussed the role of lactate as another additional biomarker and how elevated levels could indicate impaired mitochondrial function with altered energy metabolism in AD.

Response 3:

Thank you for your suggestion. We have added a section discussing lactate as a biomarker and its implications for mitochondrial dysfunction in AD. The section can be found on page [8], paragraph [1], section [2.7] in revised manuscript.

Comment 4:

Table 1 (pages 3-6). Please, the point after the reference [70] should be erased.

Response 4:

Thank you for pointing this out. We could not locate the extra point after reference in Table 1 after the reference [70].

Comment 5:

“Association of 1H MRS metabolites with biofluid biomarkers” (pages 12-14). Here, a schematic representation or an illustrative Table should be added to summarize the most relevant data found in this topic.

Response 5:

We appreciate the suggestion. Since there are only few studies which reported association of MRS metabolites with biofluid biomarkers, we provided a brief summary of these findings on page [16], paragraph [4].

Comment 6:

Future directions (pages 15-16). This section perfectly remarks the most relevant outcomes found by the authors in this work and the promising potential future action lines to pursue the topic covered in this work. No actions are requested from the authors.

Response 6:

Thank you for your positive feedback. No revisions were necessary for this section.

Reviewer 2 Report

Comments and Suggestions for Authors

The manuscript titled “Understanding Proton Magnetic Resonance Spectroscopy Neurochemical Changes using Alzheimer’s Disease Biofluid, PET, Postmortem Pathology Biomarkers and APOE Genotype” by Kara, F.; et al. is a scientific work where the authors outlined the most recent advances in the use of proton magnetic resonance spectroscopy to identify those biomarkers linked to Alzheimer’s disease and present this technique as suitable tool to monitor the progress of this malignancy. The manuscript is well-designed and the topic is of general interest. Some aspects are required to be addressed before to consider this work for further publication in the International Journal of Molecular Sciences.

1) “Alzheimer’s disease (…) accounting for 60-80 of all dementia cases. More than 55 million people live with AD and other dementias (…)” (page 1). First, the authors should add the percentage symbol after “60-80”. Then, could the authors provide quantitative data insights about the worldwide incidence of disability-adjusted life years (DALYs) according to this disease? This will greatly aid the potential readers to better understand the significance of this work.

2) “The accumulation of amyloid-beta (…) neurofibrillary tangles (…) symptoms of AD dementia manifest” (page 2). Here, the authors are mainly focused on the action of Aβ fibrils and aggregates in the progression of Alzheimer’s disease, but there exist other misfolded proteins that can play this role, such as α-syn [1] or S100A9 [2] under the effect of calcium ions.

[1] https://doi.org/10.1016/j.bbrc.2022.02.097

[2]

 https://doi.org/10.3390/biom14091091

3) “2. Commonly studied 1H MRS metabolites in AD” (pages 2-8). Here, it should be also discussed the role of lactate as other additional biomarker and how elevated levels could indicate impaired mitochondrial function with altered energy metabolism in AD.

4) Table 1 (pages 3-6). Please the point after the reference [70] should be erased.

5) “5. Association of 1H MRS metabolites with biofluid biomarkers” (pages 12-14). Here, a schematic representation or an illustrative Table should be added to summarize the most relevant data found in this topic.

6) Future directions (pages 15-16). This section perfectly remarks the most relevant outcomes found by the authors in this work and the promising potential future action lines to pursue the topic covered in this work. No actions are requested from the authors.

Author Response

(The authors gave the same response as above.)
